# Diagnostic Tools in the Detection of Physical Child Abuse: A Systematic Review

**DOI:** 10.3390/children9081257

**Published:** 2022-08-20

**Authors:** Vito Pavone, Andrea Vescio, Ludovico Lucenti, Mirko Amico, Alessia Caldaci, Xena Giada Pappalardo, Enrico Parano, Gianluca Testa

**Affiliations:** 1Department of General Surgery and Medical Surgical Specialties, Section of Orthopaedics and Traumatology, University Hospital Policlinico “Rodolico-San Marco”, University of Catania, 95123 Catania, Italy; 2Unit of Catania, Institute for Biomedical Research and Innovation (IRIB), National Council of Research, 95100 Catania, Italy

**Keywords:** child, abuse, maltreatment, neglect, physical, orthopedic

## Abstract

Child abuse is a critical social issue. The orthopedic surgeon’s role is essential in noticing signs and symptoms of physical abuse. For this reason, several authors have proposed scoring systems to identify abuse early on and reduce undiagnosed cases. The aim of this systematic review is to overview the screening tools in the literature. In 2021, three independent authors performed a systematic review of two electronic medical databases using the following inclusion criteria: physical child abuse, questionnaire, survey, score, screening tool and predictive tool. Patients who had experienced sexual abuse or emotional abuse were excluded. The risk of bias evaluation of the articles was performed according to the Newcastle–Ottawa Quality Assessment Scale Cohort Studies. Any evidence-level study reporting clinical data and dealing with a physical child abuse diagnosis tool was considered. A total of 217 articles were found. After reading the full texts and checking the reference lists, n = 12 (71,035 patients) articles were selected. A total of seven screening tools were found. However, only some of the seven diagnostic tools included demonstrated a high rate of sensitivity and specificity. The main limits of the studies were the lack of heterogeneity of evidence and samples and the lack of common assessing tools. Despite the multiplicity of questionnaires aimed at detecting validated child abuse, there was not a single worldwide questionnaire for early diagnosis. A combination of more than one test might increase the validity of the investigation.

## 1. Introduction

Child abuse is a worldwide issue, and the effects on the abused child can be both short and long term [1]. The World Health Organization recognizes child abuse and neglect as a critical international health problem [2] and defines child maltreatment as “all forms of physical and emotional ill-treatment, sexual abuse, neglect and exploitation that results in actual or potential harm to the child’s health, development or dignity” with unacceptable levels of morbidity and mortality [1,2,3,4,5]. Its real incidence is difficult to calculate and often under-reported. Despite this, on 11 December 2014, the WHO drew up an overall balance on the epidemiology of violence in all its forms with the “Report on the prevention of child maltreatment in Europe” [2] and the recent “Report on the global state on the prevention of violence 2014” [3], which reported: 852 children < 15 years of age die each year in Europe from maltreatment (the highest rate is in children under 4 years; however, Italy is in last place for the number of murders), and 44 million children are victims of physical violence (22.9%). There are four main forms of child maltreatment: neglect, physical abuse, psychological abuse and sexual abuse [2,3,4,5], and all healthcare providers should be alert to all forms of abuse, but orthopedic surgeons specifically should be focused on physical abuse, the most visible form of abuse [4]. Although cases of injuries caused by the abuse of minors are estimated to be only 1% or less of maltreated children attending the emergency room, the consequences of a missed diagnosis can have a huge influence on education, mental health, physical health and violence or criminal behavior [5]. In clinical practice, psychological development crucially depends on recognizing and diagnosing a case of abuse, both to prevent further maltreatment and to save the child’s life. It is therefore essential for physicians to have the tools available to identify the warning signs of abuse and the associated risk factors. In this regard, numerous authors have described specific clinical signs that are commonly observed in abused patients, such as head [6], chest [7], abdominal [8] trauma or trauma to the extremities [1]. Other studies have described the general characteristics of child abuse as part of a broad general analysis of injury patterns [8]. A few articles, instead, have described a discriminatory screening tool that can be used universally. The purpose of this review is to verify the work carried out by the latter authors and figure out if, currently, there are appropriate and effective scoring systems for the early identification and detection of cases of child abuse with high specificity and sensitivity in the literature.

## 2. Materials and Methods

### 2.1. Study Selection

In 2021, the following research string was used to systematically review the PubMed and Science Direct databases: “(pediatric OR child OR childhood) AND (abuse OR maltreatment OR NAT) AND (physical abuse NOT sexual abuse NOT emotional abuse) AND (screening OR diagnostic OR checklist OR questionnaire OR survey OR tool)”. Two authors (AV and MA) performed the research according to the guidelines of the Preferred Reporting Items for Systematic Reviews and Meta-Analyses (PRISMA) [9] (Appendix A). For each included original article, a standard data entry form was utilized to extract the number of patients, mean age at the time of diagnosis, sex, predictive values and year of the study.

Study quality evaluation was performed by two independent reviewers (AV and AC). Conflicts about data were resolved by consultation with a senior surgeon (VP). 

### 2.2. Inclusion and Exclusion Criteria

Eligible studies for the present systematic review included those administering a questionnaire or survey and gathering scores to make a diagnosis of physical child abuse. The initial title and abstract screening was carried out using the following inclusion criteria: physical child abuse, questionnaire, survey, score, screening tool, predictive tool. Patients who had experienced sexual abuse or emotional abuse were excluded. All remaining duplicates, articles focused on other topics and those with poor scientific methodology and no accessible abstract were excluded. 

### 2.3. Risk of Bias Assessment

The risk of bias evaluation of the articles was performed according to Newcastle–Ottawa Quality Assessment Scale Cohort Studies (NOS) [10], consisting of a three-stage assessment of the studies included. Three authors (MA, AC and GT) performed the evaluation independently. Any discrepancy was discussed with the senior investigator (VP) for the final decision. All the raters agreed on the result of every stage of the assessment.

## 3. Results

A total of n = 217 articles were found, including 3 articles added after the reference list analysis. After the exclusion of duplicates, n = 183 articles were selected. At the end of the first screening, following the previously described selection criteria, n = 28 articles were chosen for full-text reading. Ultimately, after reading the full texts and checking the reference lists, n = 12 articles were selected following the previously written criteria. A PRISMA [9] flowchart of the method of selection and screening is provided (Table 1 and Figure 1).

The following section reports the studies selected and the questionnaires and tools described.

### 3.1. Escape

One article was related to Escape. In Louwers EC et al. [11], a study that took place from July 2008 to December 2009, 18,275 children, aged up to 18 years old, were prospectively reviewed in three different German Emergency Departments (EDs). The average age was 5.5 years, and 57% of the children were male; 2.3% (n = 420) were positive for abuse, 89 patients were examined by the child abuse team, and 55 of them were classified as potential abuse cases, of which only 44 were truly positive. The sensitivity was 0.80, and the specificity was 0.98. The positive likelihood ratio was 40, and the negative likelihood ratio was 0.20. Therefore, the Escape instrument is useful in identifying children who are at high risk of abuse.

### 3.2. ISPCAN Child Abuse Screening Tools Retrospective Version (ICAST-R)

The Delphi study group developed and validated a questionnaire in seven countries [12]. A total of 842 young adults, aged 18–26 years, were examined. Internal consistency was moderate for physical abuse (Cronbach’s alpha = 0.610). 

### 3.3. Child Trauma Questionnaire (CTQ-SF)

CTQ-SF was assessed in three articles. Hernandez et al. [13] retrospectively validated the questionnaire in a Spanish-speaking population. A total of 185 abused women, aged 18-65 years, from various mental health centers, were examined. Cronbach’s α coefficient was 0.88. The confirmatory factor analysis results were: S-B χ^2^ (265) = 380.51, *p* < 0.001; S-B χ^2^/df = 1.43; comparative fit index = 0.94; root mean square error of approximation = 0.04. Bernesteid et al. [14], examined 661 individuals, including three different clinical populations. The first sample was taken from the Basel behavioral therapy clinic (n = 487). The second sample was composed of pedophilia-diagnosed patients. The third sample included patients with sleep disorders (n = 60). Cronbach’s alpha was 0.82 for physical abuse and 0.53 for physical neglect. Philip Spinhoven et al. confirmed the validity of CTQ-SF and its association with the Childhood Trauma Interview (CTI) [15]. In this study, 2308 patients, aged between 18 and 65 years, were tested. Cronbach’s alpha was good for physical abuse (0.88) and moderate for the physical neglect scale (0.60). The CTQ-SF was also mildly associated with the CTI. He et al. [16] examined the psychometric properties of the CTQ-SF in a sample of 3431 Chinese patients. Cronbach’s alpha was 0.79. Kongerslev et al. [17] and Spies et al. [18] assessed the comparative fit index for the Danish and South African populations, reporting values of 0.88 and 0.94, respectively.

### 3.4. Burn Screening

Clark et al. [19] assessed 215 patients in a prospective study carried out from April 1992 to March 1993 by introducing a list with 13 factors associated with abusive burn cases reported by an ED. Before the introduction of these factors, only 3% of the burn cases presented to the ED were reported to the social service department. Thereafter, reports rose to 12.1%.

### 3.5. Predicting Abusive Head Trauma (PredAHT) Tool

A clinical vignette study, conducted by Cowley et al. [20], analyzed the ability of the Predicting Abusive Head Trauma (PredAHT) tool to estimate the likelihood of abusive head injury (AHT) in children under 3 years of age. Twenty-five clinicians participated in the study. Clinicians expressed the probability of AHT and highlighted their child protection (CP) actions in six different clinical vignettes. The sensitivity of PredAHT was 72.3%, and the specificity was 85.7%. PredAHT significantly influenced clinicians’ probability estimates (*p* < 0.001). However, the influence of PredAHT on clinicians’ CP actions was limited. 

### 3.6. SPUTOVAMO-R

Sittig et al. [21] investigated whether a new checklist, SPUTOVAMO-R, used in emergency rooms (CHAIN-ER), was able to detect physical abuse in children in suspected cases. The study sample included 4290 children aged 0 to 7, between June 2009 and December 2010. The prevalence of physical abuse was 0.07% (95% CI 0.01 to 0.2). For every 100 suspected cases of child abuse, only 3 had been really abused (positive PV of 0.03), while 97 were not really abused (false-positive rate of 0.97 (95% CI from 0.915 to 0.904), and 0 were lost to follow-up (false-negative rate of 0.0, 95% CI 0.0 to 0.006). 

### 3.7. Screening Index for Physical Child Abuse (SIPCA)

Chang et al. [22] examined the effectiveness of the screening test SIPCA for the physical abuse of children. Children aged up to 14 were included for analysis. A database of 1961 hospitals in 17 cities was used (n = 58,558). A SIPCA score of 3 had a sensitivity of 86.6% and a specificity of 80.5%, and a SIPCA score of 4 had a specificity of 93.1%, but had a lower sensitivity of 71.8.

## 4. Discussion

Child abuse can lead to developmental issues, adverse physical and psychological effects, including subsequent ill health, higher rates of chronic conditions, high-risk health behaviors and a shortened lifespan. Child maltreatment is a social issue that needs to be diagnosed early and eradicated.

Several authors have proposed scoring systems useful in the detection of physical maltreatment. In our systematic review, seven diagnostic tools were included. Good predictive values were found for Escape, CTQ-SF and SIPCA, but we did not find a questionnaire that surpassed the others in terms of effectiveness and diagnostic accuracy. One of them, SPUTOVAMO-R, had a high probability of identifying a non-abused child as positive, and it is not recommended. Lastly, although most of the questionnaires obtained high rates of sensitivity and specificity, some limitations reduce their potential daily clinical use. 

The Escape screening instrument, described by Louwers et al. [11] in 2012, consists of a six-item checklist addressing risk factors for child abuse. It had been suggested for every child ED case; nevertheless, it was not developed as an injuries checklist. In the selected study, it obtained a high rate of sensitivity and specificity. At the same time, a limitation of this study is that the real rate of child abuse is not available because potential abuse could have distorted the result. Chang at al. [22] analyzed a database of 1961 hospitals in 17 cities (n = 58,558) and tested a new questionnaire, SIPCA. The scores depend on logistic regression models, on age and on patterns of injuries. The study showed a high rate of sensitivity and specificity for a SIPCA score of 3 and a higher sensitivity and a lower specificity for a SIPCA score of 4. Therefore, a score of 3 on SIPCA represents a good compromise in terms of sensitivity and specificity in detecting physical abuse and is an excellent threshold for placing abuse in differential diagnosis. However, positive or negative results must be further investigated by clinical and structural tests. A limitation of this study could be the truthfulness of the data in hospital records; however, the consistency of the results in the hospitals and in several states confirms the quality of the study. Another questionnaire, CTQ-SF, has been evaluated in three studies. It includes 28 questions. In 2012, Hernandez et al. [13] retrospectively validated the questionnaire in a Spanish-speaking population. The Cronbach’s α coefficient obtained was satisfactory. However, the results obtained in this study cannot be extended to the non-clinical or male population, and the retrospective data could be distorted. In 2014, Bernesteid et al. [15] examined the German version of CTQ-SF. Cronbach’s alpha was very high for physical abuse and moderate for physical neglect, but several limitations of this study need to be considered: there are no means to verify the answers and the use of a mainly clinical sample. The last study examined the tool with good results [13] and obtained a good Cronbach’s alpha for physical abuse and a moderate value for the physical neglect scale. However, patients can have flawed memories in the reconstruction of the abuse. The ICAST-R [12] questionnaire, tested by Dunne et al. in 2005, comprises 15 questions and has been translated into six languages (Arabic, Hindi, Malay, Marathi, Russian and Spanish). A total of 842 young adults (120 in Russia, 89 in Egypt, 120 in Lebanon, 124 in India, 125 in Malaysia, 120 in Colombia and 144 in Kyrgyzstan) took part in the study. Cronbach’s alpha was moderate for physical abuse. However, the samples of subjects examined by country were small and unrepresentative and therefore could not be extended with certainty to the entire population. Additionally, gender differences were missing in the questionnaire. Another weakness of this study is that the concurrent validity and reliability of test–retest have not been examined. The burn screening, a checklist with 13 factors associated with burns [19], has increased the number of new diagnoses of abuse and the effective referral to social service. However, Clark et al. analyzed few cases. In 2015, Cowley et al. [20] examined a clinical vignette study, PredAHT. The six parameters analyzed in the predictive head injury clinical prediction tool were: head or neck bruising, seizures, apnea, rib fracture, long-bone fracture and retinal hemorrhage. It was introduced to help clinicians to decide when further diagnosis should be required in the event of a head injury in children under the age of three. PredAHT improved the probability estimates but had a minimal impact on physician actions. Since vignettes differ from real situations, studies on vignettes are often criticized because of potential limits of external validity, and they are often subjected to evaluation error by the clinicians, who are also influenced by distractors. Finally, from 2009 to 2010, Sittig et al. [21] examined a new questionnaire, SPUTOVAMO-R. It includes six questions. However, this study shows that this checklist gives a high rate of false positives, 97%, which could lead to unnecessary treatment, and as panel members rely on subjective information in the patient file, such as risk factor assessments, we cannot exclude a certain level of implicit bias.

## 5. Conclusions

Despite the multiplicity of questionnaires aimed at detecting validated child abuse, there is not a single worldwide questionnaire for the early diagnosis. A combination of more than one test might increase the validity of the investigation, but there is a high risk of these evaluations taking a long time, and they are not always available in clinical practice.

Further studies may be necessary to evaluate the accuracy and the reliability of the questionnaires available in the literature and to find other diagnostic tools helpful for the physician and most importantly for the child.

Nowadays, physical child abuse is still a social issue, being underestimated by professionals, mainly due to the lack of recognition resulting from not yet completely reliable diagnostic tools.

## Figures and Tables

**Figure 1 children-09-01257-f001:**
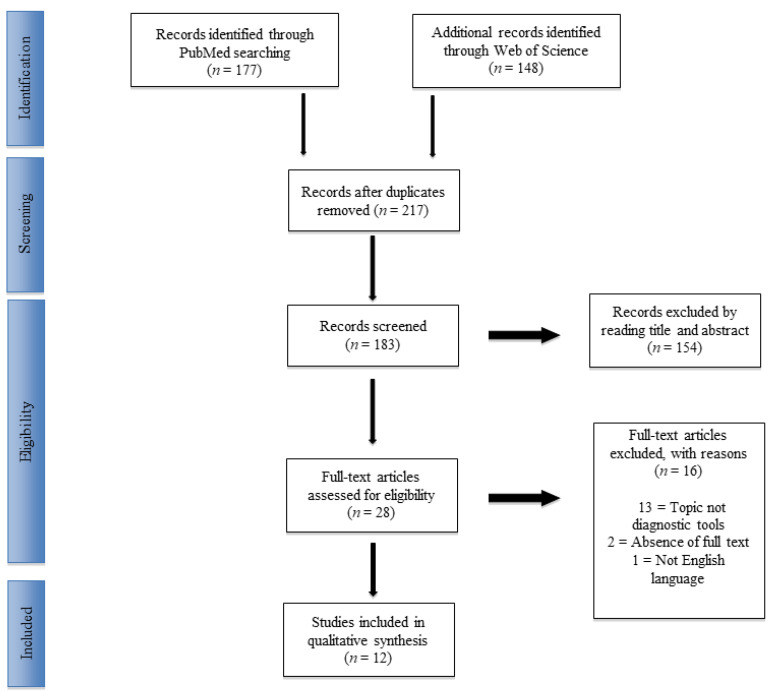
PRISMA (Preferred Reporting Items for Systematic Reviews and Meta-Analysis) flowchart of the systematic literature review.

**Table 1 children-09-01257-t001:** Results of individual studies collected from the literature.

Author	Samples	Intervention	Outcome Measures	Results	Limits of the Study
Louwers et al., 2014	89 children (55 were potential abuse cases, 44 were really positive) Average age was 5.5, 57% were male, 44 were positive	They prospectively examined children for child abuse with ESCAPE questionnaire	Sensitivity, specificity, +LR, −LR	The sensitivity was 0.80, the specificity was 0.98, +LR was 40, −LR was 0.20 Therefore, the escape instrument is useful for identifying children who are at high risk of abuse	The real rate of child abuse is not available because potential abuse could distort the end result
Dunne et al., 2009	842 children (18–26 years)	Delphy group developed and validated a questionnaire for child abuse in 7 countries, ICAST-R	Cronbach’s alpha	Cronbach’s alpha was moderate, 0.610, for physical abuse	The sample was small, Gender differenceswere missing, The validity and reliability have not been verified
Hernandez et al., 2013	185 women abused (18–65 years)	They validated the Spanish version of CTQ-SF questionnaire, retrospectively, for child abuse	Cronbach’s alpha S-B χ^2^, *p*, S-B χ^2^/df, CFI, RMS	S-B χ^2^ (265) = 380.51, *p* < 0.001,…; S-B χ^2^/df = 1.43 CFI = 0.94, RMS = 0.04, Cronbach’s alpha was 0.88 Cronbach’s α coefficients obtained were really satisfactory	The results obtained cannot be extended to the non-clinical or male population, and the data could be distorted
Bernesteid et al., 2014	661 individuals divided into 3 different clinical populations	They validated the German version of CTQ-SF questionnaire for child abuse	Cronbach’s alpha	Cronbach’s alpha was 0.82 for physical abuse and 0.53 for physical neglect This study shows a good validity of the German model with the exception of the physical neglect scale	There are no means to verify the answers. The use of a mainly clinical sample is another limitation of the present study. Finally, it would have been useful to check whether the results could be generalized to non-Swiss German-speaking people
Spinhoven et al., 2014	2308 patients (18–65 years)	They verified the validity of CTQ-SF and its association with the CTI	Cronbach’s alpha	Cronbach’s alpha was good for physical abuse (0.88) and moderate for the physical neglect scale (0.60) The results on validity and reliability available make the CTQ-SF a valid tool for the screening of various forms of abuse	In this study there may have been flawed memories in the reconstruction of the abuse by the patient
He et al., 2019	3431 (1943 men; 1488 women)	They examined the psychometric properties of the 28-item CTQ-SF in a Chinese population	Cronbach’s alpha	CTQ-SF total was 0.79; emotional neglect 0.76; physical neglect 0.52; Emotional abuse 0.68; Physical abuse 0.72; Sexual abuse 0.77	Lack of diversity in the selection of subjects. Horizontal study.
Kongerslev et al., 2019	142; 68% women;	They evaluated the psychometric properties of the Danish CTQ-SF in a clinical sample.	Comparative fit index	CTQ-SF total was 0.878; emotional neglect 0.62; physical neglect 0.48; emotional abuse 0.62; physical abuse 0.37; sexual abuse 0.93	A small subsample of adult outpatients diagnosed with personality disorders. Horizontal study.
Spies et al., 2019	314 women (170 HIV uninfected; 144 HIV infected)	They evaluated the psychometric properties of the South Africa CTQ-SF in a clinical sample.	Comparative fit index; goodness of fit index	Comparative fit index 0.94; goodness of fit index 0.85	Lack of diversity in the selection of subjects. Horizontal study.
Clark et al., 1997	215 patients	They prospectively validated the effectiveness of a new list with 13 factors associated with abusive burns	Number of new diagnoses of child abuse to ED	Reports rose to 12.1%	A limitation of this study is the few cases used
Sittig et al., 2016	4290 children (0–7 years)	They investigated a new questionnaire, SPUTOVAMO-R, for child abuse	Prevalence, positive PV, false-positive rate, false-negative rate	The physical abuse’s prevalence was 0.07% (95% CI 0.01 to 0.2). For every 100 cases of suspected child abuse, only 3 were really abused (positive PV of 0.03); however, 97 were not really abused (false-positive rate of 0.97 (95% CI from 0.915 to 0.904), and 0 were lost to follow-up (false-negative rate of 0.0, 95% CI 0.0 to 0.006(8). This study shows that this questionnaire gives a high rate of false positives, 97%, which could lead to unnecessary treatment	A certain level of implicit bias cannot be excluded
Chang et al., 2005	58,558 children In 17 cities	They examined the effectiveness of the SIPCA questionnaire for child abuse	Sensitivity, specificity	A SIPCA score of 3 had a sensitivity of 86.6% and a specificity of 80.5%, a SPICA score of 4 had a specificity of 93.1%, but had a lower sensitivity of 71.8. The study shows a high rate of sensitivity and specificity for a SPICA score of 3 and a higher sensitivity and a lower specificity for a SPICA score of 4. Therefore, a score of 3 on SIPCA represents a good compromise	The truthfulness of the data in hospital records
Cowley et al., 2018	Children under 3 years age	A clinical vignette study analyzing the Predicting Abusive Head Trauma (PredAHT) tool to estimate the likelihood of abusive head injury (AHT)	Sensitivity, specificity	The sensitivity of PredAHT was 72.3% and the specificity was 85.7%.PredAHT significantly influenced clinicians’ probability estimates (*p* < 0.001)	Since vignettes differ from real situations, studies on vignettes are often criticized because of the potential limits of external validity and their propensity for evaluation error by the clinicians, who are also influenced by distractors

## Data Availability

Not applicable.

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
