# Peer review of "Diagnostic Tools in the Detection of Physical Child Abuse: A Systematic Review"

_children, 2022, doi:10.3390/children9081257_

Round 1

Reviewer 1 Report

I appreciate the efforts of the authors but I regret to find this revision entirely non-responsive. 

Again, the paper makes the erroneous statement (starting in the Abstract) that physical abuse is “the most frequent form of maltreatment”—this is patently untrue. They expressly indicate in the Abstract that this is a “systematic review”, despite the title indicating a “scoping review”—again confusing the two. By expressly limiting their database and excluding (using the NOT term) any other form of co-occurring maltreatment, the authors are again omitting relevant literature. The authors respond that they wanted to only focus on PubMed and Web of Science but I cannot see why they would purposely exclude literature conducted by nurses, social workers, and psychologists (an anti-multidisciplinary stance) who work in this field regularly and often generate substantial literature in this domain. The authors are still confusing reliability with validity (when they purport to be interested in sensitivity and specificity)—why bother including reliability at all? Again, the actual inclusion of irrelevant (retrospective) papers suggests a failure to appreciate the point of the task—to identify child abuse while it is happening and determine whether there are tools to detect this accurately.

I would implore the authors to please consider the consequences of publishing work that may have arrived at erroneous conclusions about such an important topic. Not only does such research damage the entire scientific enterprise (and thereby undermine public confidence in science writ large), but with a topic of such major societal and public health importance, arriving at unsupported conclusions risks harming the very children that the authors clearly want to benefit—victims of physical abuse. Even publishing *potentially* flawed findings misleads readers about what is expected of replicable, rigorous research. If the authors had conducted a more thorough review of the literature and come to the conclusion that there was insufficient sensitivity and accuracy, that could motivate the field to do better. Unfortunately, this paper does not accomplish this without serious reservations.

Language issues remain. The citations numbers do not align at all with the actual content of the sentences using those citations.

Author Response

Thank you for your revision. Here you find our response point-by-point

  1. Thank you for your comment. The uppercases and lowercases have been revised according to MDPI format.
  2. Thank you for your comment. Quantitative results added in the abstract.
  3. Thank you for your comment. Keywords are now in alphabetic order.
  4. Thank you for your comment. We added keywords.
  5. Thank you for your comment. The introduction has been improved.
  6. Thank you for your comment. The novel of this article is the inconsistency between the data displayed here and the data in the previous literature. In previous studies, there was a correlation between visual impairment and fractures. Here, there is not.
  7. Thank you for your comment. Surgical treatment has been included in the introduction, surgical treatment of hip fracture has been described in the discussion, and we added citations too.
  8. Thank you for your comment. The paragraph to explain the biases of previous studies has been added.
  9. Thank you for your comment. An illustrative figure of the study has been added to the materials and methods.
  10. Thank you for your comment. The article has been rewritten in some parts to make everything more understandable.
  11. Thank you for your comment. The paragraphs have been rearranged.
  12. Thank you for your comment. The conclusion has been rewritten.
  13. Thank you for your comment. Further studies needed has been added.
  14. Thank you for your comment. We added statistical data.
  15. Thank you for your comment. The JCM template has been followed.
  16. Thank you for your comment. The English language has been proofread.

Reviewer 2 Report

The modifications improve the comprension of the work

Author Response

thank you for your comments

Reviewer 3 Report

The authors used one method of reviewing the research literature to understand the precision of tools to detect child physical abuse- PRISMA.  Out of the 183 articles that possibly utilized tools to detect child physical abuse, only 28 were promising enough to review and only 12 fit the inclusion criteria for full critique.  The authors described each study in a table and critiqued each study in the text of the paper.  While seven tools were detected through the review and a few actually were used to examine child physical abuse with children under the age of 18 (mostly below age 5), none were deemed good enough to endorse.  

I always worry that relevant studies are thrown out by only reading the title and abstract or that the key words miss potential studies for inclusion. I am not an expert in this literature so I don't know if that was the case.  I know PRISMA is a standardized method of review, but I have found that not all studies choose keywords wisely.  When my doctoral students do reviews, I have them not only read the key studies that seem to hit on the topic head on, but also read studies referenced in those papers. Only when they have exhausted those avenues do I feel confident they have identified most of a literature to date. But, that is not in this protocol and someone who works in the area of diagnostic tools for child physical abuse will need to verify that the authors did not miss an important contribution to the literature.

This type of review is important for the field to illuminate gaps in the literature. The potential for other scholars to test the most promising tools more rigorously or to adapt the tools to improve their predictive validity or to develop new tools with better predictive validity is high. 

An editor will need to go through the manuscript and make some small changes to make the text more fluid since the authors presumably do not speak English as their first language, but the dysfluencies are minor and can be corrected. 

Author Response

Thank you for your comments. Here you will find a point-by-point responde:

  • The introduction has been changed by explaining why the data in the present study is different.
  • We added the use of odds ratio to implement our statistics.
  • The general statement has been substituted with a statement backed by literature.
  •  We added statistical data. The factors mentioned in the discussion are there to widen it and give a better picture of the situation. The study focuses only on visual impairment.

This manuscript is a resubmission of an earlier submission. The following is a list of the peer review reports and author responses from that submission.

Round 1

Reviewer 1 Report

Children #1483726

“Diagnostic tools in the detection of physical child abuse: A scoping review”

This manuscript purports to conduct a review of the literature to identify potential screening tools for detecting physical child abuse—a worthwhile topic.  However, fatal methodological problems compromise the paper.

First critical problem is the approach to searching the literature. Because the authors restricted their database to two medical databases (unnecessarily restrictive) and restricted the search terms to exclude papers that also include other forms of maltreatment, they excluded papers that consider multiple forms of maltreatment inclusive of physical abuse. As a result, they are implying that some measures have been inadequately considered when there are, for example, 21 papers alone simply using the search term Child Trauma Questionnaire-Short Form in a single database, or 9 with the ICAST. The actual search process is flawed in other ways, but there is simply no way that only 214 articles would have been identified on first search.

The second critical problem is an evident misunderstanding of what constitutes an appropriate outcome for accuracy as they are confusing and confounding reliability and validity—fundamentally distinct psychometric characteristics. The authors are clearly interested in accuracy but appear not to recognize the difference in foundational psychometric qualities to accurately judge the merits of a measure with regard to its potential accuracy. They thus do not appear to understand how to judge a measure accurately even if the actual purpose of a paper was not to demonstrate accuracy; consequently, I am dubious they accurately reviewed the individual papers to extract the necessary information.

The third critical problem is an inconsistency of what they purport their goal to be (to identify physical abuse while it is occurring) and the literature they extracted. Several of the papers they report are from adults’ retrospective report and do not even involve children.  There is ample literature on assessment of children for child abuse that would not rely on such retrospective reports.

This is all apart from several fundamental and serious errors in characterizing the condition of child maltreatment.  For example, the most common form of maltreatment is unquestionably neglect (not, as they suggest, physical abuse)—there is no source in the world that would dispute this. Further, educational personnel, followed by legal personnel, are those most likely to report maltreatment (not as they suggest, medical personnel). This suggests a failure to appreciate the current state of the literature on child maltreatment.

Additionally, there are pervasive problems throughout with the authors’ presentation. First, the authors do not appear to appreciate the difference between a “scoping review” and a “systematic review” which they seem to consider interchangeable; consult Pham et al., 2014 and/or Munn et al., 2018 for guidance. Second, the citations do not correspond to the content provided in text with multiple errors in trying to convey the scope of maltreatment and its impact (apart from reporting author names incorrectly)—appropriate citations and referencing is required to avoid plagiarism and ensure accuracy. And there are abundant instances in inaccurate, incorrect, and unclear language choice throughout the paper accompanied by actual errors (e.g., incomplete sentences, suggesting they selected 7 studies but reporting 9), culminating in a single paragraph that is over a page long.

Collectively, this paper suffers from grievous methodological problems that would require a complete re-analysis and reconceptualization of how to identify accuracy of the available measures.

Reviewer 2 Report

This paper was well written and the organization was exemplary. I find few faults with it and recommend acceptance with no revision. 

Reviewer 3 Report

 This is a really relevant work that rigorously analyzes the studies carried out on different diagnostic tools for child maltreatment, particularly physical maltreatment.
The text is well structured although the introduction could have provided more updated WHO data on the prevalence of child maltreatment. Data from 2013 are provided when WHO has more updated reports.
The methodology followed in the systematic review is correct and follows the PRISMA model.
The results are very valuable for practice as they conclude which tools have the best predictive capacity for child physical abuse and which are not recommended for use.
Although it is not the direct objective of the article, the conclusions on strategies to improve diagnosis in clinical practice, which are really agile, are lacking.